

SciPost Phys. 1(1), 005 (2016)

# Exactly solvable quantum few-body systems associated with the symmetries of the three-dimensional and four-dimensional icosahedra

**T. Scoquart[1,2], J. J. Seaward[2*], S. G. Jackson[3] and M. Olshanii[2]**

**1** Département de Physique, Ecole Normale Supérieure, 24, rue Lhomond, 75005 Paris, France
**2** Department of Physics, University of Massachusetts Boston, Boston, MA 02125, USA
**3** Department of Mathematics, University of Massachusetts Boston, Boston Massachusetts 02125, USA

* Joseph.Seaward001@umb.edu

## Abstract

The purpose of this article is to demonstrate that non-crystallographic reflection groups can be used to build new solvable quantum particle systems. We explicitly construct a one-parametric family of solvable four-body systems on a line, related to the symmetry of a regular icosahedron: in two distinct limiting cases the system is constrained to a half-line. We repeat the program for a 600-cell, a four-dimensional generalization of the regular three-dimensional icosahedron.


## 1 Introduction

The language of Lie groups that is traditionally employed when constructing new integrable quantum few- and many-body systems ( [1]; [2], Ch. 5 therein; [3]) inadvertently prohibits

non-crystallographic symmetries from being considered, since no associated Lie groups exist. Some additional consistency can be gained if the context is shifted away from Lie groups and towards discrete reflection groups, affine or finite, classic or exceptional, crystallographic or not. In this paper, we explicitly construct two quantum solvable four- and five-body systems based on the non-crystallographic groups $H_3$ and $H_4$ respectively.

We build on the general results obtained in the course of work devoted to extending the realm of integrable systems to the cases covered by the exceptional reflection groups [4] (the case of $\tilde{F}_4$ in particular), long thought to be irrelevant (see [2], paragraph 5.2.3(c) therein): prior to [4], the scope of applicable refection groups has been limited to $A_{N-1}$ (respectively $\tilde{A}_{N-1}$) and $C_N$ (resp. $\tilde{C}_N$) [1,5–7]. These groups correspond to $N$ atoms of the same mass, on a line (resp. ring) and on a half-line (resp. in a box) respectively.

The essence of the extension presented in [4] is a diversification of the variety of maps between the particle coordinates and the Cartesian spaces in which the reflection groups operate. Such improvement allowed one to include ensembles of particles of different mass in consideration. As a result, it was possible to devise a general scheme according to which every reflection group—finite or affine—whose Coxeter diagram [8] does not have forks, generates an exactly solvable quantum few- or many-body system (or a few-parametric family of them) of hard-core particles on a line, a half-line, or in a box or on a ring. We should note that when the associated reflection group is known, construction of the particle eigenstates *per se* follows a known scheme that exists for any solvable kaleidoscopic cavity with homogeneous Robin boundary conditions, irrespectively of whether it has a particle analogue or not [1,2,9–14].

Regretfully, the above scheme does not allow for any extension to the case of finite strength interactions, if one requires the interactions be both of a two-body nature and act only on a contact. The physical reason is that the for finite interactions, particles are allowed to explore the whole multidimensional coordinate space where the reflection group operates. However, with the exception of the group $A_{N-1}$ (and $C_N$ with restrictions), the number of mirrors in the group grows much faster then the number of particle pairs. In Section. 4, we treat this phenomenon in more detail.

## 2 $H_3$: symmetries of an icosahedron

Consider four hard-core particles on a line, with masses $m_1$, $m_2$, $m_3$, $m_4$, and coordinates $x_1$, $x_2$, $x_3$, $x_4$ respectively, with $x_1 < x_2 < x_3 < x_4$. A coordinate transformation $x_i = \sqrt{\mu/m_i} z_i$ reduces the system to a four-dimensional particle of mass $\mu$. The arbitrary mass scale $\mu$ is distinct from the total mass and can be chosen at will. The particle will be moving inside a hard-walled wedge formed by three hyperplanes of particle-particle contact with the outer normals

$$\boldsymbol{\alpha}_i = \sqrt{m_i/(m_{i-1}+m_i)}\boldsymbol{e}_{i-1} - \sqrt{m_{i-1}/(m_{i-1}+m_i)}\boldsymbol{e}_i$$
$$\text{for } i = 2, 3, 4 , \tag{1}$$

with $\boldsymbol{e}_i$ being unit vectors along the $z_i$-axes. The mutual orientation of the planes is not generic: these three planes cross along a line oriented along a unit vector $\boldsymbol{e}_{\text{COM}} \equiv \sum_{i=1}^{4} \sqrt{m_i/M} \boldsymbol{e}_i$, where $M \equiv \sum_{i=1}^{4} m_i$ is the total mass. Projection of the radius vector $\boldsymbol{z} \equiv (z_1, z_2, z_3, z_4)^\top$ onto the direction $\boldsymbol{e}_{\text{COM}}$ is proportional to the position of the center of mass in the physical coordinates: $\boldsymbol{e}_{\text{COM}} \cdot \boldsymbol{z} = \sqrt{M/\mu} \sum_{i=1}^{4} m_i x_i / M$. For any set of masses, the time evolution of the center of mass coordinate $X_{\text{COM}} \equiv \sqrt{\mu/M} \boldsymbol{e}_{\text{COM}} \cdot \boldsymbol{z}$ can be separated from the rest of the dynamics.

Dihedral angles between the plane of contact between $i-1$'st and $i$'th particles and its

 SciPost Phys. 1(1), 005 (2016)

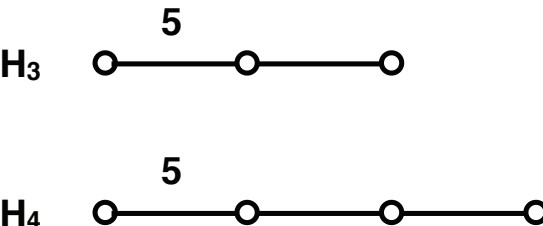

Figure 1: Coxeter diagrams [8] corresponding to the reflection groups $H_3$, i.e. the symmetry group of a regular icosahedron, and $H_4$, i.e. the symmetry group of a 600-cell, the four-dimensional cousin of a regular icosahedron. The way the diagrams encode the relative orientation of the generating mirrors of the group is described both in the main text and in the caption to Fig. 3.

analogue for $i$'th and $i + 1$'st particles are given by [7]

$$\theta_{(i-1)\,i\,(i+1)} = \arctan \sqrt{\frac{m_i(m_{i-1} + m_i + m_{i+1})}{m_{i-1}m_{i+1}}} \ . \tag{2}$$

For two non-overlapping pairs, $m_{i-1}$-$m_i$ and $m_j$-$m_{j+1}$ with $j > i$, the corresponding hyperplanes are orthogonal to each other. Consider a full set of the particle-particle mirrors (three, for four particles). Some mirror arrangements form kaleidoscopes: in this case, the transformations of space caused by chains of sequential reflections form a finite group[1]. A complete list of these instances exists [15, 16], and it is proven to be complete. Each instance of a kaleidoscopic mirror arrangement is encoded by a Coxeter diagram [8]. Fig. 1 provides examples of Coxeter diagrams for the reflection groups $H_3$ and $H_4$. Vertices correspond to the generating mirrors. Two vertices not connected by an edge correspond to two mirrors at a right angle between them. Two vertices connected by an unmarked edge give two mirrors at 60° between them. Finally, edges labeled with an index $n$ produce a pair of mirrors at $(180/n)°$.

According to the rules presented immediately above, the $H_3$ diagram at Fig. 1 produces three mirrors at angles 36°, 60°, and 90° between them. It is easy to verify that each member of the following two-parametric family of the mass spectra produces such a set of particle-particle hyperplanes:

$$\begin{cases} m_1 = \frac{\xi+1}{(5-2\sqrt{5})\xi-1}\,m_2 & \text{with } \frac{1}{5-2\sqrt{5}} \leq \xi \leq 3 \ , \\ m_3 = \xi\,m_2 & \text{and } x_1 < x_2 < x_3 < x_4 \ . \\ m_4 = \frac{\xi(\xi+1)}{3-\xi}\,m_2 \ , \end{cases} \tag{3}$$

The family is parametrized by an overall mass scale $m_2 \geq 0$ and a dimensionless parameter $\xi$. The reason for the bounds on $\xi$ is the additional requirement of non-negativity of the masses involved. Two limiting cases deserve special attention, $\xi \to (5-2\sqrt{5})^{-1} + 0$ and $\xi \to 3-0$. In the first limit, the leftmost mass $m_1$ diverges. In a frame with the origin coinciding with the mass $m_1$ and co-moving with with it, the problem reduces to a one-parametric family of three-body problems on a right half-line:

$$\begin{cases} m_3 = \frac{1}{5-2\sqrt{5}}\,m_2 & \text{with } m_1 \to +\infty, \ x_1 = 0, \\ m_4 = \left(\frac{5}{2} + \frac{11}{2\sqrt{5}}\right)m_2 \ , & \text{and } 0 < x_2 < x_3 < x_4 \ . \end{cases}$$

_______________

[1]In the case of particles on a ring or in a hard-wall box, the group is countably infinite.

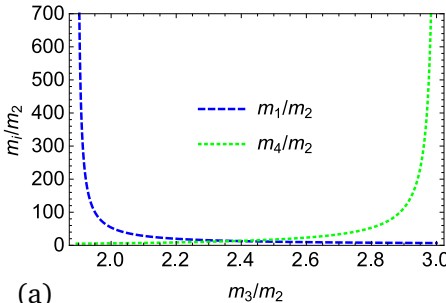
(a)

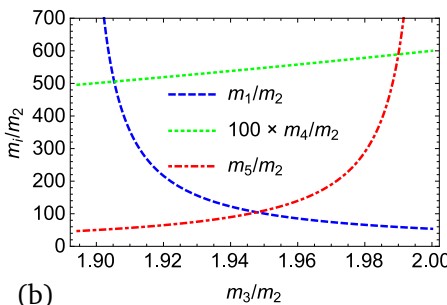
(b)

Figure 2: Mass ratios leading to icosahedral symmetries. $m_i$ is the mass of the $i$th particle in the sequence. (a) A family of four-body mass spectra that realize the reflection group $H_3$. (b) A family of five-body mass spectra that realize the reflection group $H_4$.

In the second limit, the rightmost mass $m_4$ diverges. Here, we obtain a one-parametric family of problems on a left half-line:

$$\begin{cases} m_1 = \frac{2}{7-3\sqrt{5}}\, m_2 & \text{with } m_4 \to +\infty,\ x_4 = 0, \\ m_3 = 3\, m_2\,, & \text{and } x_1 < x_2 < x_3 < 0\,. \end{cases}$$

Fig. 2(a) illustrates the dependencies (3).

When the hyperplanes of the particle-particle contact form a kaleidoscopic cavity, construction of the eigenstates becomes an easy task. In the case of $H_3$—the full symmetry group of an icosahedron—the sequential applications of reflections about the three generating mirrors (1) produce 120 orthogonal transformations $\hat{g}$ that form this group. Eigenstates of the Hamiltonian, all of which are scattering states, are parametrized by an incident wavevector $\boldsymbol{k}$:

$$\boldsymbol{\alpha}_i \cdot \boldsymbol{k} > 0\,, \text{ for } i = 2, 3, 4\ :$$

the corresponding Bethe Ansatz eigenstates [1, 2, 9–14] have a form

$$\psi_{\boldsymbol{k}}(\boldsymbol{z}) = \text{const} \times \sum_{\hat{g}} (-1)^{\mathscr{P}(\hat{g})} \exp[i(\hat{g}\boldsymbol{k}) \cdot \boldsymbol{z}]\,. \tag{4}$$

Here, $\mathscr{P}(\hat{g})$ is the parity of the group element $\hat{g}$: the parity of the number reflections about the generating mirrors (1) that lead to this element.

For the problems with no bound states, scattering states of zero energy become the most fundamental object of interest. In the case of Bethe Ansatz states based on kaleidoscopic symmetries, the pure reflection members of the reflection group—that also contains rotations and combinations of a rotation and a reflection—play the central role. The group $H_3$ contains 15 pure reflections, that correspond to 15 symmetry planes of a regular icosahedron. Let $\boldsymbol{\beta}$ be one of the 15 normals to the corresponding mirrors, where we assume, in order to avoid ambiguities, that $\boldsymbol{\alpha}_i \cdot \boldsymbol{\beta} > 0$, for all $i = 1, 2, 3$ and all 15 normals $\boldsymbol{\beta}$ [2]. The normals $\boldsymbol{\beta}$ can be obtained by sequential applications of reflections about the generating mirrors to a normal $\boldsymbol{\alpha}$ to one of them. It can be easily shown that the lowest degree anti-invariant polynomials of the corresponding group [15],

$$\psi_{\boldsymbol{k}=0}(\boldsymbol{z}) = \text{const} \times \prod_{\beta} (\boldsymbol{\beta} \cdot \boldsymbol{z})\,, \tag{5}$$

[2]Remark that according to an established terminology, the vectors opposite to $\boldsymbol{\alpha}_i$ and $\boldsymbol{\beta}_j$, i.e. $-\boldsymbol{\alpha}_i$ and $-\boldsymbol{\beta}_j$, are the *simple roots* and the *positive roots* respectively.

produce the desired zero-energy eigenstates of the problem[3]. Fig. 3 illustrates the probability density in the state (5). Here, the position of the center of mass, $X_{\text{COM}} \equiv \sqrt{\mu/M}(e_{\text{COM}} \cdot z)$, is set to zero. In the residual three-dimensional subspace of the space the $z$ coordinates belong to, the state (5) factorizes into a product of a function of the radial coordinate $r = |z - (e_{\text{COM}} \cdot z)e_{\text{COM}}|$ (that is proportional to $r^{15}$ in the $H_3$ case) and a function of angular coordinates. It is the latter that is shown at Fig. 3.

## 3 $H_4$: symmetries of a 600-cell

$H_4$ is the full symmetry group of a 600-cell [8], a regular four-dimensional polytope (a four-dimensional Platonic solid) that constitutes a four-dimensional analogue of a regular three-dimensional icosahedron. Its three-dimensional "surface" consists of regular tetrahedra, five meeting at each edge.

In the case of the $H_4$ reflection group, one more mirror, at 60° to the mirror corresponding to the rightmost vertex of the $H_3$ diagram (Fig. 1) is added. Accordingly, a fifth particle is added to the system. The resulting two-parametric family of mass spectra is

$$
\begin{cases}
m_1 = \frac{\xi+1}{(5-2\sqrt{5})\xi-1}\, m_2 & \\
m_3 = \xi\, m_2 & \text{with } \frac{1}{5-2\sqrt{5}} \leq \xi \leq 2 \,, \\
m_4 = \frac{\xi(\xi+1)}{3-\xi}\, m_2 & \text{and } x_1 < x_2 < x_3 < x_4 < x_5 \,, \\
m_5 = \frac{\xi(\xi+1)}{(3-\xi)(2-\xi)}\, m_2 \,, &
\end{cases}
\tag{6}
$$

where the two governing parameters are again a mass scale $m_2$ and a dimensionless ratio $\xi \equiv m_3/m_2$. This family is illustrated at Fig. 2(b).

Here, like in the $H_3$ case, we have two nontrivial special points. At $\xi \to (5-2\sqrt{5})^{-1} + 0$, the mass spectrum converges to

$$
\begin{cases}
m_3 = \frac{1}{5-2\sqrt{5}}\, m_2 & \\
m_4 = \left(\frac{5}{2} + \frac{11}{2\sqrt{5}}\right) m_2 & \text{with } m_1 \to +\infty,\ x_1 = 0, \\
m_5 = \left(\frac{47+21\sqrt{5}}{2}\right) m_2 \,, & \text{and } 0 < x_2 < x_3 < x_4 < x_5 \,,
\end{cases}
$$

and the system reduces to a four-body problem on a right half-line. Here, we are again assuming a moving frame whose origin coincides with the coordinate of the infinitely massive first particle at all times.

The limit $\xi \to 2 - 0$ leads to a four-body problem on the left half-line:

$$
\begin{cases}
m_1 = (27 + 12\sqrt{5})\, m_2 & \text{with } m_5 \to +\infty,\ x_5 = 0, \\
m_3 = 2\, m_2 & \text{and } x_1 < x_2 < x_3 < x_4 < 0 \,. \\
m_4 = 6\, m_2 \,, &
\end{cases}
$$

For each member of the $H_4$ family of mass spectra (6), the outer normals to the generating mirrors are given by

$$
\alpha_i = \sqrt{m_i/(m_{i-1}+m_i)}\, e_{i-1} - \sqrt{m_{i-1}/(m_{i-1}+m_i)}\, e_i
$$
$$
\text{for } i = 2, 3, 4, 5 \,,
\tag{7}
$$

---

[3]Since the Laplacian commutes with each element of the group, it would take the lowest degree homogeneous anti-invariant polynomial (5) to a homogeneous anti-invariant polynomial of two degrees lower. However, by construction, there is no such polynomial [15]. Thus, the action of the Laplacian must produce zero, i.e. (5) must be a zero energy eigenstate of it.

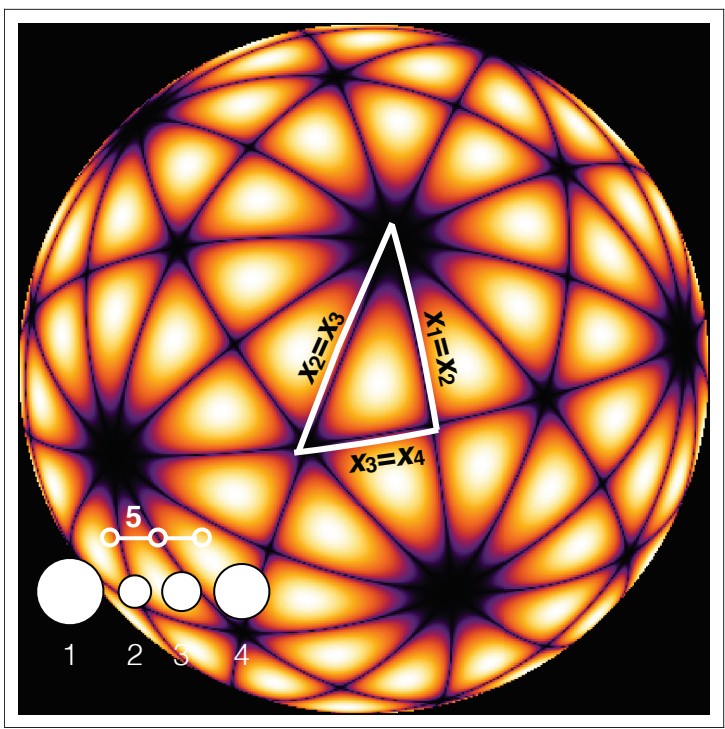

Figure 3: The white triangle bounds the physically allowed values of particle positions. In it, we show the angular distribution, in the space of relative motion, of the probability density in the zero-energy state (5) of four hard-core particles on a line, with a mass spectrum belonging to the family (3). Note that in this state, the angular distribution does not depend on the radial coordinate. A smooth continuation of this state to the remainder of the sphere is also shown, to illustrate the symmetry of the state. It is evident that the three angles of the "physical" triangle are 36°, 60°, and 90°. These values are encoded in the Coxeter diagram (lower left corner) as the index 5 (as in 36° = $\pi/5$) above the edge between the leftmost and the middle vertices, "empty" index between the middle and the rightmost vertices (the "3" in $\pi/3$ is omitted by convention,) and an "empty" edge between the leftmost and the rightmost vertices for the right angle (also omitted by convention.) Vertices themselves correspond to the sides of the triangle. From the particle perspective (labels and relative masses indicated below the Coxeter graph,) vertices of the Coxeter graph represent pairs of consecutive particles while the edges and the indices above them correspond the the consecutive particle triplets and the mass ratios in the triplet whose ratios are governed by (2).

Eigenstates of the $H_4$ problem will contain many more plane waves than in the $H_3$ case. All possible sequences of reflections about the generating mirrors for the full symmetry group of the 600-cell, that contains 14400 orthogonal transformations $\hat{g}$. Exactly like in the three-dimensional case, none of the transformations affects the dynamics of the center-of-mass coordinate $X_{COM} \equiv \sqrt{\mu/M}(e_{COM} \cdot z)$, with $e_{COM} \equiv \sum_{i=1}^{5} \sqrt{m_i/M} e_i$ being the corresponding unit vector and $M \equiv \sum_{i=1}^{5} m_i$ being the total mass. There are 60 pure reflections in the $H_4$, with the corresponding normals $\beta$. The generic scattering states are given by the general formula (4), with the sum running over all 14400 elements of the group, and $\alpha_i \cdot k > 0$, for $i = 2, 3, 4, 5$. The zero-energy scattering state will be again given by the expression (5), where the product consists of 60 factors, and $\alpha_i \cdot \beta > 0$, for all $i = 1, 2, 3, 4$ and all 60 $\beta$'s, to avoid ambiguity.

## 4 Summary and outlook

In this paper, we propose two new families of exactly solvable quantum four- and five-body problems; these cases are associated with the symmetries of an icosahedron and a 600-cell (i.e. a four-dimensional analogue of an icosahedron) respectively. This result explicitly demonstrates that non-crystallographic reflection groups can be used to construct quantum integrable few-body systems, on par with the crystallographic ones. In addition to the generic eigenstates we also analyze the zero-energy eigenstates, that correspond to the lowest-degree anti-invariant polynomials of the corresponding reflection group.

We believe our results can not be extended to the case of finite-strength $\delta$-interactions between the particles if the local (i.e. contact) two-body nature of the interactions is to be preserved. Indeed it can be shown that, in order to preserve integrability of the system, the $\delta$-function potentials at the 15 mirrors of the $H_3$ group must have the same strength, infinite or finite. In the finite case, any permutation of the four particles involved is possible, leading to 6 hyperplanes of contact, a number that differs from the number of mirrors in the group. In contrast, for a given permutation, preserved over time, of the hard-core particles, only 3 hyperplanes of contact are physically accessible; the number of mirrors accessible, given the particles' impenetrability, is also 3. Likewise, in the case of the $H_4$ group, the number of the hyperplanes of particle-particle contact (i.e. 10) would differ from the number of the mirrors (i.e. 60).

The relevance of the "counting" argument above can be strengthened by the most tangible case of two $\delta$-interacting particles in the field of a fixed $\delta$-potential of a different strength. Here, in the two-dimensional plane of system's coordinate space, the potential is localized along the horizontal, vertical, and one of the diagonal lines, a set that is clearly not closed under reflections about its own members. And, as it is shown in [18], the eigenstates show features inconsistent with integrability, diffraction being the primary one. The mirror symmetry ($C_2$ in this case) and the associated integrability could be restored by adding an unphysical interaction that acts when the particles are located at the same distance from the potential but on the opposite sides of it. And finally the system can be returned to the realm of physical by raising the strength of the stationary potential to infinity, while keeping the "unphyisical" part of the interparticle interactions. For an initial condition where both particles start at the same side of the potential, they would simply not be capable to explore the "unphyisical" part. In this example, both the empirical relevance and the integrability of a model can be preserved, but only at the expense of reducing choice of one of the interactions to infinite values.

The remaining non-crystallographic reflection groups, $I_2(m)$, associated with the symmetries of regular polygons, deserve attention. Even though the resulting three body integrable systems—whose classical versions were analysed in [17]—are conceptually much simpler then most of the other problems of this class, there are two aspects that call for closer consideration.

Firstly, as it has been shown classically in [17], a many-body system that contains integrable few-body sub-systems shows a slowdown of relaxation: a quantum version of the phenomenon is in order. The case of $I_2(m)$ symmetry is the most empirically relevant, since it can be realized with *only two* atomic species. Secondly, exact eigenstates, albeit not of the Bethe Ansatz type, can be obtained for any set of masses of three hard-core particles on a line; a separation of the radial and angular components of the relative motion can be used. This case can be used to analyze the relationship between the Bethe Ansatz integrability (along with possible associated Liouville integrability [4]) and the existence of the exact solutions in general.

On a different front, the answer to the question of existence of particle realizations of reflection symmetries with the bifurcating Coxeter diagrams, $D_n(\tilde{D}_n)$, $\tilde{B}_n$, and $E_{6,7,8}(\tilde{E}_{6,7,8})$, remains as elusive as ever.

# Acknowledgements

The authors thank Vanja Dunjko for help and comments.

**Funding information**  This work was supported by the US National Science Foundation Grant No. PHY-1402249, the Office of Naval Research Grant N00014-12-1-0400, and a grant from the *Institut Francilien de Recherche sur les Atomes Froids* (IFRAF). Financial support for TS provided by the Ecole Normale Supérieure is also appreciated.

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
