# Peer review of "Exactly solvable quantum few-body systems associated with the symmetries of the three-dimensional and four-dimensional icosahedra"

_SciPost Physics, doi:SciPost Phys. 1, 005 (2016)_

## Round 1 · Referee Report · Anonymous · 2016-9-13

Strengths
1- An new way of looking at Guadin's system associated with the finite Coxeter groups H_3, H_4 in the *impenetrable* regime, namely as few-body particle systems with different masses.
Weaknesses
1- Only hard-core particles are considered. It is a natural question whether Gaudin's system [1] associated with H_3 and H_4 in the *interacting* regime can also be interpreted as a few-body system.
Report
Gaudin observed in a landmark paper that the quantum boson system of $n$ particles with delta-potential interaction admits natural generalisation in terms of finite Coxeter groups [1]. Furthermore, the generalised systems from [1] also admits generalisations in terms of affine Weyl groups [1,2, 8,9,10,11,12,13].
For classical Coxeter groups these systems have natural particle interpretations. For example, the original n-particle model with pairwise interaction corresponds to the permutation group $S_n$ (on the line) or affine $S_n$ (on the circle). The other classical Coxeter groups appear when restricting the particles to a half-line or interval, or by distributing them symmetrically around the origin.
Recently the last two authors showed that the system associated with the affine Weyl group $\tilde F_4$ in the impenetrable regime (i.e. when the interaction goes to +infinity) can be mathematically identified with a system of four one-dimensional hard-core particles with mass ratios 6, 2, 1, and 3 in a hard-wall box.
The preprint under review is a continuation of the above idea, but now applied to the finite non-crystallographic Coxeter groups of type H_3 and H_4. The paper can be summarised as follows: In the impenetrable regime Gaudin's systems [1] associated with H_3 and H_4 can be mathematically identified with a system of four (for H_3) or five (for H_4) one-dimensional hard-core particles on a line, if the ratios of the masses m_j/m_2 are certain explicit rational functions of a dimensionless parameter \xi. For very special values of $\xi$ the system reduces to a system of three (for H_3) or four (for H_4) one-dimensional hard-core particles on a half-line.
Requested changes
1- In the paper "Yang's system of particles and Hecke algebras", Heckman and Opdam relate the spectral problem for the system in [1] to the representation theory of the graded Hecke algebra of the corresponding Coxeter group. Furthermore, they also prove the crucial Plancherel formula (in the repulsive and attractive regime). This important paper should be cited.
2- Typo: Page 6, figure 3: morion --> motion
3- Discuss possible extension to interacting case (sea Weakness)
Author: Maxim Olshanii on 2016-09-13 [id 55]
(in reply to Report 1 on 2016-09-13)First of all, we thank the referee for a truly thorough reading of our manuscript and for s/his suggestions. We are currently
preparing a new version to be posted at the arXiv
Changes:
(1) We will add the suggested reference, it is indeed very relevant;
(2) We will correct the typo;
(3) We will add a note on finite interaction extensions to the Outlook section.
Concerning (3), a comment is in order:
Prior to our $\tilde{F}_{4}$ paper, even among the four classic reflection groups, only two were producing \emph{local two-body} interaction
implementations, and only one of them unconditionally.
(*) $A_{N-1}$($\tilde{A}_{N-1}$) corresponds to the relative motion of $N$ $\delta$-interacting particles on a line(ring), with any value of the
interaction strength allowed;
(*) $B_{N}=C_{N}$($\tilde{C}_{N}$) corresponds to $N$ $\delta$-interacting particles on a half-line(box) bounded by one(two)
$\delta$-barriers of an \emph{infinite} strength.
Already for $B_{N}=C_{N}$, one can see containing the requirement for having only local two-body interactions is. If one replaces an
infinite barrier by a finite one, integrability will still be preserved. However, in that case, one would need to allow some particles to venture
on the other side of the barrier. And this is where they will nonlocal (albeit still two-body) interactions: a particle a distance $l$ from the
barrier, will interact with another particle also at a distance $l$ from the barrier but \emph{on the other side} of it. Formally, these nonlocal
interactions are present even in the case of an infinite barrier: but particles can be contained indefinitely on one side of the barrier, and in this
case, the non-physical interactions do not lead to any measurable consequences.
For other reflection groups, the situation is even more severe, and we currently believe that no finite-strengh
(at least for some pairs of particles) particle implementations of reflection groups, with local two-body interactions, exists,
besides the $A_{N-1}$ and $C_{N}$. In the paper, we give a simple counting argument for why the $H_{3}$ group can not
produce a physically sound finite-strength-interacting system.

---

## Round 2 · Referee Report · Anonymous · 2016-10-3

Strengths
1. Finding specific mass ratios in the few-body hard-core models with particle-particle contact interaction for which one obtains H_3 and H_4 type Lieb-Liniger models with infinite delta interaction strenghts by a change of coordinates.
2. Obtaining an explicit wave expansion of the solutions of the associated spectral problem.
Weaknesses
1. Identification of Lieb-Liniger models associated to exceptional reflection groups and few-body models
can only be done for infinite delta-interaction strengths (hard-core particles). A precise analysis what prevents extension to finite delta-interaction strengths is missing.
2. The ideas in this paper have been worked out before for reflection group of type F_4 in ref. [4] of the paper under review. The paper is a rather straightforward exercise to adjust the techniques to reflection
groups of type H_3 and H_4.
Report
The Lieb-Liniger Bose gas on the line is a famous integrable one-dimensional many body system naturally attached to the symmetric group. It admits a generalisation as integrable system in which the role of the symmetric group is taken over by a finite reflection group. The particle-particle contact interaction of the Lieb-Liniger Bose gas is replaced by delta-interactions at the root hyperplanes of the reflection group.
For classical Weyl groups there still is a reasonable interpretation as a one-dimensional many body system. For other types, in case of infinite delta-interaction strengths, the model can sometimes admit an interpretation as a hard-core few-body model on the line with distinguishable particles.
The paper under review gives an example of this phenomenon.
The idea that one-dimensional particles with different masses and with particle-particle contact interaction can be related by a (nonorthogonal) change of coordinates to a model describing equal particles but with delta-potential interactions along "nonphysical" hyperplanes goes back to McGuire in 1963 (it is reference [7] in the paper under review). The main point of the paper is to show that if the masses of the particles are exactly such that the hyperplanes x_i=x_{i+1} turn into the simple root hyperplanes for some nonclassical finite reflection group, then the model is equivalent to the associated generalised Lieb-Liniger model with infinite delta-interaction strengths. In ref. [4] of the paper under review this was already worked out in case of the finite reflection group of type F_4 and its affine version. In the present paper the same techniques are used for the noncrystallographic reflection groups H_3 and H_4.
Requested changes
1. See item 1 at weaknesses. A thorough analysis of what goes wrong for finite interaction strengths should be added.
Small comments:
2. plains --> planes (several times).
3. Section 2: Formula for e_{COM}\cdot\mathbf{z} is wrong (only so for \mu=M).
4. "m_{i-1}-m_i and m_i-m_{i+1} planes" need explanation.
5. I do not understand why giving an arctan-formula for the angle, instead of the (more standard) way of expressing the angle using arccos (which does not make the mass-dependence more difficult).
6. Page 3: Ref. [15] should be [16] I suppose.
Author: Maxim Olshanii on 2016-10-09 [id 59]
(in reply to Report 1 on 2016-10-03)We thank the referee for their careful reading and insightful critique.
>> 1. See item 1 at weaknesses. A thorough analysis of what goes wrong for finite
>>interaction strengths should be added.
We added an explanatory paragraph in the conclusion section.
>> Small comments:
>> 2. plains --> planes (several times).
Corrected
>> 3. Section 2: Formula for e_{COM}\cdot\mathbf{z} is wrong (only so for \mu=M).
Corrected
>> 4. "m_{i-1}-m_i and m_i-m_{i+1} planes" need explanation.
Corrected
>> 5. I do not understand why giving an arctan-formula for the angle, instead of the (more >> standard) way of expressing the angle using arccos (which does not make the mass-
>> dependence more difficult).
The convention is consistent with the well accepted convention used in the
foundational McGuire article and the rest of our work in this topic.
>> 6. Page 3: Ref. [15] should be [16] I suppose.
Corrected

---

## Round 2 · List of Changes

1. Added a reference to Heckman and Opdam's "Yang’s system of particles and Hecke algebras."
2. Added a second paragraph to the concluding section about extending the results to finite-strength delta-potentials.
3. Minor grammatical, spelling and word-choice changes.

You are currently on this page

---

## Round 4 · List of Changes

1.) Paragraph added to introduction and conclusion about the difficulty extending these solutions to finite delta-interaction potentials.

2.) Clarified labeling convention on the formula governing the mass ratios.

3.) Formula in section 2 was formerly true only for when \mu=M.

4.) Correction of reference pointer on page three and minor spelling corrections

---

## Editorial Decision

published